# Higher Omega-3 Index Is Associated with Better Asthma Control and Lower Medication Dose: A Cross-Sectional Study

**DOI:** 10.3390/nu12010074

**Published:** 2019-12-27

**Authors:** Isobel Stoodley, Manohar Garg, Hayley Scott, Lesley Macdonald-Wicks, Bronwyn Berthon, Lisa Wood

**Affiliations:** 1Priority Research Centre for Healthy Lungs, Hunter Medical Research Institute, Newcastle, NSW 2305, Australia; Isobel.Stoodley@uon.edu.au (I.S.); Hayley.Scott@newcastle.edu.au (H.S.); Bronwyn.Berthon@newcastle.edu.au (B.B.); 2School of Health Sciences, University of Newcastle, Callaghan, NSW 2308, Australia; Lesley.Wicks@newcastle.edu.au; 3School of Biomedical Science and Pharmacy, University of Newcastle, Callaghan, NSW 2308, Australia; Manohar.Garg@newcastle.edu.au

**Keywords:** omega-3 index, asthma, inflammation, fatty acids, nutritional biomarkers

## Abstract

Asthma is a chronic inflammatory airway disease, associated with systemic inflammation. Omega-3 polyunsaturated fatty acids (n-3 PUFA) have established anti-inflammatory effects, thus having potential as an adjunct therapy in asthma. This study aimed to compare erythrocyte n-3 PUFA in adults with (*n* = 255) and without (*n* = 137) asthma and determine the relationship between erythrocyte n-3 PUFA and clinical asthma outcomes. Subjects had blood collected, lung function measured and Juniper Asthma Control Questionnaire (ACQ) score calculated. Fatty acids were measured in erythrocyte membranes by gas chromatography, and the omega-3 index (O3I) was calculated (% eicosapentaenoic acid + % docosahexaenoic acid). O3I was similar in subjects with and without asthma (*p* = 0.089). A higher O3I was observed in subjects with controlled or partially controlled asthma (ACQ < 1.5) compared to subjects with uncontrolled asthma (ACQ ≥ 1.5) (6.0% (5.4–7.2) versus 5.6% (4.6–6.4) *p* = 0.033). Subjects with a high O3I (≥8%) had a lower maintenance dose of inhaled corticosteroids (ICS) compared to those with a low O3I (<8%) (1000 μg (400–1000) versus 1000 μg (500–2000) *p* = 0.019). This study demonstrates that a higher O3I is associated with better asthma control and with lower ICS dose, suggesting that a higher erythrocyte n-3 PUFA level may have a role in asthma management.

## 1. Introduction

Asthma is a chronic inflammatory disease of the airways, affecting 2.5 million Australians in 2014–2015 [1]. Globally, it is estimated that 334 million people have asthma [2]. Airway inflammation in asthma is triggered by exposures such as allergens and viruses, and causes airway hyper-responsiveness (AHR), airway smooth muscle contraction and excess mucous production [3,4]. This results in the hallmark symptoms of asthma including breathlessness, wheezing, chest tightness and persistent cough [2]. Systemic inflammation is also a feature of asthma, with circulating C-reactive protein (CRP) levels shown to be elevated in people with asthma [4,5,6], which is associated with poorer lung function and more severe airway inflammation [6].

Current treatment for asthma predominantly involves inhaled corticosteroid (ICS) medication, which helps to control symptoms and exacerbations and to improve lung function and quality of life by reducing airway inflammation [7,8]. However, dietary patterns are also being investigated for their potential preventative or therapeutic role. It has been suggested that a Western dietary pattern, high in energy, saturated fats, sugars and salt, may increase the prevalence and severity of asthma, independent of socioeconomic and lifestyle factors [9]. Additionally, a Mediterranean dietary pattern, which is nutrient dense and high in fish, fruit and vegetables, could be protective, reducing the incidence and the severity of asthma symptoms [9].

One such way that the Mediterranean diet might be effective in reducing asthma symptoms is due to the high intake of omega-3 polyunsaturated fatty acids (n-3 PUFA). n-3 PUFA include eicosapentaenoic acid (EPA, C20:5n-3), docosapentaenoic acid (DPA, C22:5n-3) and docosahexaenoic acid (DHA, C22:6n-3) and are commonly found in significant amounts in marine sources such as salmon, herring and sardines [10]. These fatty acids have been found to inhibit inflammatory processes within the body, with benefits in cardiovascular disease well established [10]. These include suppression of transcription factors that control the production of circulating inflammatory cytokines CRP, tumour necrosis factor alpha (TNF-α), interleukin (IL)-1β and IL-6 [11,12]. n-3 PUFA also compete with omega-6 polyunsaturated fatty acids (n-6 PUFA), resulting in the downregulation of arachidonic acid-derived immune and inflammatory mediators including 3- and 5-series prostaglandins, thromboxanes, leukotrienes and lipoxins [3,13]. These same pathways and inflammatory mediators are involved in AHR in people with asthma [11,12]. Hence, it is possible that n-3 PUFA could play a role in the prevention or treatment of asthma.

Various studies have investigated n-3 PUFA in asthma. Mouse models have shown that increased DHA intake is associated with reduced eosinophil infiltration into the lungs [13] and that by increasing the ratio of n-3:n-6 in lung tissue, interleukins can be downregulated [14]. In humans, research is conflicting. Some studies have found that supplementation with n-3 PUFA can decrease inflammatory markers and improve asthma symptoms [15,16], while others found no changes to AHR or airway inflammation [17]. A Cochrane meta-analysis conducted in 2000, and updated in 2011 with nil changes, including 9 randomised control trials (RCTs) of both adults and children, concluded that there was no benefit or risk for the use of dietary marine fatty acids in people with asthma [18,19]. Furthermore the European Academy of Allergy and Clinical Immunology have released a position statement emphasizing that until more standardized trials with assessment of pre-intervention fatty acid levels have been conducted, there is no recommendation for n-3 PUFA in asthma and other allergic diseases [20].

The conflicting evidence highlights a need for more research in this area. Hence, this study aimed to examine the relationship between n-3 PUFA status and clinical outcomes in Australian adults with asthma. Firstly, it is unclear whether n-3 PUFA status is impaired in Australian subjects with asthma compared to health controls, thus an aim of this project was to investigate and describe the differences between these two groups. We hypothesized that individuals with asthma would have poorer n-3 PUFA status compared to those without asthma. Furthermore we hypothesized that subjects with asthma and a high n-3 PUFA status would have better clinical outcomes than those with low n-3 PUFA status. These aims were examined using the omega-3 index (O3I), which has been validated as a reliable measure of dietary n-3 PUFA intake and reflects long-term n-3 PUFA status [21,22]. O3I is the sum of erythrocyte EPA and DHA, expressed as a percentage of total erythrocyte membrane fatty acids [23]. A secondary aim of this project was to examine the effects of obesity on O3I in adults with asthma. Obesity in asthma is associated with poorer asthma control, greater severity, higher medication doses and more frequent exacerbations than healthy weight individuals [24]. One mechanism suggested to underpin this relationship is the chronic low-grade inflammation associated with obesity [25,26]. Considering the anti-inflammatory properties attributed to n-3 PUFA, it is possible that n-3 PUFA may attenuate this inflammation. Whether these interactions exist in obese asthmatic subjects is unknown. Therefore, we hypothesized that in an obese asthmatic population, those with a lower O3I would have poorer clinical and biochemical outcomes compared to those with a higher O3I.

## 2. Materials and Methods

### 2.1. Subjects

Subjects were pooled from seven previously published research studies [25,27,28,29,30,31]. Subjects were adults (≥18 years of age) with (*n* = 255) and without (*n* = 137) asthma, recruited at the Hunter Medical Research Institute (HMRI), NSW Australia, from existing research volunteer databases or by media release. Subjects were nonsmoking (never smoked, or ceased at least 6 months prior). Asthma was defined as a doctor’s diagnosis of asthma with documented history of AHR. All asthmatic subjects were classified as stable with no asthma exacerbation, respiratory tract infection or oral corticosteroid use in the preceding four weeks. Exclusions included current smokers, use of systemic anti-inflammatory or immunosuppressant medications or current cancer diagnosis. All studies were conducted at the Hunter Medical Research Institute, Newcastle, Australia between 2006–2015. All procedures involving human subjects were approved by the Hunter New England Human Research Ethics Committee (Ethics approval numbers: 11/06/15/3.03; 14/02/19/3.01; 13/07/17/4.03; 08/10/15/5.07; 09/03/18/5.05; 05/03/09/3.09; 09/05/20/5.07). All subjects provided written informed consent.

Clinical assessment and blood collection were performed during a single clinic visit. Subjects underwent spirometry including forced expiratory volume in one second (FEV_1_) and forced vital capacity (FVC) (Koko, nSpire Health, Longmont, CO, USA or Medgraphics, PFS/D and BreezeSuite software; Medgraphics, Saint Paul, MN, USA) in accordance with American Thoracic Society and European Respiratory Society guidelines [32,33]. All asthmatic subjects completed the six-item Juniper Asthma Control Questionnaire (ACQ6) [34]. Partially or well controlled asthma was defined as ACQ < 1.5, while uncontrolled asthma was defined as ≥1.5 [34]. Clinical asthma pattern was determined according to Global Initiative for Asthma (GINA) recommendations [8]. Maintenance ICS doses were recorded and converted to beclomethasone equivalents. Body weight was measured in 0.1 kg increments using calibrated electronic scales (Nuweigh EB8271; Newcastle Weighing Services, Wickham, Australia). Height was calculated to the nearest millimetre using a wall-mounted stadiometer (Seca 220; Seca, Hamburg, Germany). Body mass index (BMI) was calculated as body weight (kg)/height (m)^2^.

### 2.2. Sputum Induction and Analysis

Sputum induction and bronchial provocation were performed using 4.5% hypertonic saline over 15.5-min nebuliser time [35]. Lower respiratory sputum portions were selected and dispersed using dithiothreitol. Total cell counts and cell viability (trypan blue exclusion) were determined from cytospins.

### 2.3. Plasma Inflammatory Markers

Venous blood was collected after a 12 h overnight fast. Commercial ELISAs were used to determine plasma high sensitivity CRP (MP-Biomedicals, Orangeburg, NY, USA), IL-6 and TNF-α (R&D Systems, Minneapolic, MN, USA), according to manufacturer’s instructions.

### 2.4. Erythrocyte Membrane Fatty Acid Preparation

Whole blood was collected in EDTA tubes and centrifuged at 3000× *g* at 4 °C for 10 min. Red blood cells were separated and stored at −70 °C before analysis.

After thawing, the erythrocytes were lysed, and their membranes solubilised and purified using the method described by Tomoda et al. [36]. Then, 12 mL of hypotonic tris buffer (10 mM tris hydroxymethyamino methane/5 mM ascorbate buffer, pH 7.4) was added to approximately 500 μL of erythrocytes and vortexed. After standing on ice for five minutes, 12 mL of 0.25 M glucose solution was added. The sample was vortexed, stood on ice for another five minutes, then centrifuged at 10,000 rpm at 4 °C for 10 min. The supernatant was discarded and the procedure repeated twice more (resuspending the pellet by vortexing) using the same quantities of tris and glucose solutions above, but centrifuging at 12,000 rpm at 4 °C for 10 min and then 15,000 rpm at 4 °C for 20 min. The pellet was then resuspended in approximately 250 μL each of tris and glucose solutions and stored at −20 °C prior to analysing for fatty acid content.

### 2.5. Fatty Acid Determination

Total erythrocyte fatty acids were determined using the method established by Lepage and Roy [37]. Here, 2 mL of a methanol/toluene mixture (4:1 *v*/*v*), containing C21:0 (0.02 g/L) as internal standard and BHT (0.12 g/L), was added to 200 μL of erythrocyte membrane suspension. Fatty acids were methylated by adding 200 μL acetyl chloride dropwise while vortexing and heating to 100 °C for one hour. After cooling, the reaction was stopped by adding 5 mL 6% K_2_CO_3_. The sample was centrifuged at 3000 rpm at 4 °C for 10 min to facilitate separation of layers. The upper toluene layer was used for gas chromatography analysis of the fatty acid methyl esters, using a 30 m × 0.25 m (DB-225) fused carbon-silica column coated with cyanopropylphenyl (J & W Scientific, Folsom, CA, USA). Both injector and detector port temperatures were set at 250 °C. The oven temperature was 170 °C for two minutes, increased 10 °C/min to 190 °C, held for one minute, then increased 3 °C/min up to 220 °C and maintained to give a total run time of 30 min. A split ratio of 10:1 and an injection volume of 5 mL was used. The chromatograph was equipped with a flame ionisation detector, autosampler and autodetector. Sample fatty acid methyl ester peaks were identified by comparing their retention times with those of standard mixture of fatty acid methyl esters and quantified using a Hewlett Packard 6890 Series Gas Chromatograph with Chemstations Version A.04.02.

### 2.6. Fatty Acid Calculations

Saturated, monounsaturated, polyunsaturated, n-3 PUFA and n-6 PUFA are reported as a percentage of total fatty acids. Omega-3 Index is calculated as ((erythrocyte membrane EPA (mg) + erythrocyte membrane DHA (mg))/total erythrocyte fatty acids (mg)) × 100 [22]. Harris et al. [22] classify an O3I between 0–4% as undesirable, 4–8% as intermediate and ≥8% as desirable, for cardioprotective benefits. An O3I cutpoint of ≥8% was used for the current analysis to represent a high O3I. The omega-6:omega-3 ratio was determined by dividing total omega-6 fatty acids (%) by total omega-3 fatty acids (%).

### 2.7. Statistical Analysis

Data are reported as median (interquartile range (IQR)) for nonparametric data or mean ± standard deviation (SD) for parametric data. Data were analysed using GraphPad Prism 7.0 for Windows (GraphPad Software, La Jolla, California, USA) and STATA 15 (StataCorp, College Station, TX, USA). Comparisons for continuous data were performed using either the unpaired *t*-test or Mann–Whitney test for nonparametric variables. Logistic regression analysis was used, adjusted for age, BMI and sex, for comparisons between subjects with versus without asthma; subjects with asthma with well or partially controlled asthma versus poorly controlled asthma; and subjects with asthma with low versus high O3I. Age and gender adjusted two-factor ANOVA was used to analyse the interaction between obesity and O3I status on clinical and biochemical asthma outcomes. Chi-squared testing was used for categorical variables. In this study, *p*-values < 0.05 were considered statistically significant.

## 3. Results

### 3.1. Comparison of Nonasthmatic and Asthmatic Subjects

Table 1 shows the clinical characteristics of subjects with and without asthma. There was no significant difference in age between the groups, however the BMI was higher and there were fewer females in the nonasthmatic group. After adjusting for age, gender and BMI, lung function was lower in the asthmatic population. CRP was significantly higher in the asthma population, however there was no significant difference in the other systemic inflammatory markers. Subjects without asthma had a significantly lower percentage of eosinophils in sputum, higher percentage of macrophages in sputum, higher percentage of SFAs, a lower percentage of MUFAs and omega-3 PUFA and a higher ratio of n-6:n-3 fatty acids.

### 3.2. Asthma Clinical Markers, Systemic Inflammation and O3I

While O3I did not differ between subjects with and without asthma, when comparing within the asthma group alone, subjects with partially controlled or well controlled asthma (ACQ6 < 1.5) had a significantly higher O3I compared to those with uncontrolled asthma (ACQ6 ≥ 1.5) (6.0% (5.4–7.2) versus 5.6% (4.6–6.4), respectively, *p* = 0.033) (Figure 1).

When analysing clinical asthma measures, there was no significant difference in lung function or asthma severity in subjects with a higher (≥8%) versus lower (<8%) O3I (Table 2). However subjects with higher O3I had a significantly lower range of maintenance ICS dose (beclomethasone equivalents) (Table 2, Figure 2). There were no significant differences in systemic inflammatory markers between higher and lower O3I after adjusting for age, gender and BMI (Table 2).

### 3.3. Asthma, O3I and Obesity

Due to the interesting nexus of asthma and obesity, we compared clinical asthma markers and systemic inflammation in subjects with asthma divided into obese (BMI ≥ 30 kg/m^2^) and nonobese (BMI < 30 kg/m^2^) groups with high (≥8%) and low (<8%) O3I (Table 3). The analysis was adjusted for gender and age. We found that subjects who were obese with a lower O3I had a significantly higher range of maintenance ICS medication dose compared with obese subjects with a higher O3I (*p* = 0.0002) (Table 3). While there was significance difference detected in all lung function measures, only FEV1/FVC ratio revealed an obesity interaction. Both obesity and O3I were significant predictors of CRP, but not for TNF-α or Il-6. Obesity and O3I were not significant predictors for ACQ.

## 4. Discussion

To the best of our knowledge, this is the first study reporting that a lower omega-3 index is associated with poorer asthma control in adults with asthma. Additionally, a higher O3I was associated with a lower maintenance ICS dose. Interestingly, this was most significant in the subjects who were also obese, showing a similar dose range of maintenance ICS to nonobese subjects with asthma. Considering the high medication burden and reduced quality of life in people with asthma, our study suggests that higher levels of n-3 PUFA could be utilised as an adjunct therapy in the treatment of asthma.

Our first aim was to investigate the differences in erythrocyte fatty acid levels and O3I between subjects with and without asthma in an Australian population. Contrary to our hypothesis there was no difference in O3I, subjects with asthma had a better fatty acid profile, with lower saturated fatty acids, higher monounsaturated and n-3 fatty acids and a lower n-6 PUFA to n-3 PUFA ratio. This is in contrast to the results of Zhou et al. [38], who found that subjects in China with asthma had a fatty acid profile composed mostly of SFAs, while those without asthma contained more PUFAs. Similar to our study, supplement and dietary intake data were not available to determine whether these differences were reflective of different dietary or supplement patterns or an effect of asthma. In severe asthma, dysregulation of lipid metabolism pathways has been observed; in particular, n-3 PUFA pathways are impaired, while n-6 PUFA pathways remain unaffected [39]. This may explain the differences between the two studies, as our population had mild to moderate asthma while Zhou et al. did not present data on the severity in their population and potentially had more severe asthma phenotypes. Furthermore, the subjects with asthma were older than the control group (mean 58 years old versus 25 years old), and without adjusting for this may have been a confounder explaining the differences between the two groups.

This study demonstrated that a lower omega-3 index is associated with poorer asthma control in adults. Subjects with uncontrolled asthma had a significantly lower O3I than those with well controlled or partially controlled asthma. Our findings are supported by a cross-sectional study conducted in 2011, which observed a significant positive relationship between EPA and DHA consumption (measured using a Food Frequency Questionnaire (FFQ)) with asthma control and lung function (FEV_1_) [40]. Our study reinforces this finding, using the objective and longer-term measure of O3I, which provides stronger evidence for this relationship.

Another significant finding was the relationship between O3I and maintenance ICS dose. Subjects with a higher O3I had a significantly lower range of maintenance ICS dose. A similar finding has been shown in a recent RCT by Papamichael et al. [41], where children with asthma were prescribed a Mediterranean diet supplemented with two meals of 150 g cooked fatty fish per week for 6 months, compared to a control group following their normal diet. They found that while there was no difference between lung function, asthma control and quality of life scores, there was a significant reduction in medication use for children in the intervention group [41]. In a study examining exercise-induced bronchoconstriction, n-3 PUFA supplementation (3200 mg EPA + 200 mg DHA for eight weeks) has also been demonstrated to reduce bronchodilator use in adults [42]. Given irregular reporting of respiratory outcomes such as ICS dose and asthma control in the available literature, systematic reviews on n-3 PUFA in asthma have highlighted the need for more high quality research in this area [19,43]. In particular, the relationship between n-3 PUFA and medication use would be of great interest for future research, given our results. Our findings suggest that achieving an O3I ≥ 8% could be a beneficial target for people with asthma in order to reduce maintenance ICS dose. This target also corresponds with cardioprotective recommendations [22]. Nutritionally, to achieve this O3I, it would equate to consuming ≥800 mg EPA and DHA per day, or 4–5 serves of mostly oily fish per week [44]. Intervention studies are required to confirm our observations, particularly to elucidate the ideal dose and duration needed to achieve this status in an asthmatic population, as well as the most effective pathway (supplementation versus whole foods).

As obesity is generally associated with increased asthma severity, poorer asthma control and more frequent exacerbations, it was important to analyse the relationship between O3I, obesity and asthma outcomes [24]. When we examined obese and nonobese asthmatics according to O3I, obese asthmatics with a higher O3I had a lower range of maintenance ICS doses compared with obese asthmatics with a lower O3I. This is particularly important, as obesity is associated with a reduced response to ICS medication, requiring higher doses to achieve protective effects [45]. Our findings suggest that omega-3 fatty acids could be a potential nonpharmacological approach to assist in the management of asthma, however our findings require confirmation by intervention studies. A recent study by Lang et al. found that supplementing with 4 g/day of fish oil over 24 weeks did not affect asthma control, lung function, exacerbations or have any impact on medication use in overweight and obese adolescents and young adults with asthma [46]. This study did not investigate inflammation. Rather, it reported fatty acid status within inflammatory cells such as monocytes and granulocytes, so without confirmation that the supplementation was reducing inflammatory pathways it may be possible that the dose or length of treatment was not sufficient to affect clinical changes in asthma.

Interestingly, we did not find any differences in systemic inflammatory markers between asthmatics with lower and higher O3I, after adjusting for age, gender and BMI. This has been confirmed in other studies investigating the association between n-3 PUFA and inflammatory markers in asthma [47,48,49]. However, one study used a semiquantitative FFQ rather than objectively measuring n-3 PUFA [47]. Another study trialed a fish oil supplement for eight weeks (800 mg or 3400 mg per day), with neither dose producing a reduction in CRP [48].

However, other studies have found positive effects of n-3 PUFA on systemic inflammatory markers in asthma. Farjadian et al. [15] studied children with asthma and demonstrated a reduction in TNF-α and IL-17A in 72% of subjects after n-3 PUFA supplementation (180 mg EPA and 120 mg DHA daily) for 3 months [15]. This study also demonstrated an improvement in asthma symptoms, while Mickleborough et al. [16] found that supplementing n-3 PUFA (3.2 g EPA and 2.2 g DHA) in elite athletes over three weeks suppressed exercise-induced bronchoconstriction and inflammatory markers such as TNF-α and IL-1β.

A review examining the impact of n-3 PUFA supplementation on inflammatory biomarkers across a variety of diseases found that, particularly in cardiac populations and the critically ill, omega-3 fatty acid supplementation can reduce a variety of inflammatory biomarkers including CRP, IL-6 and TNF-α [50]. However systematic inflammation is not reported in systematic reviews on n-3 PUFA and asthma due to irregularity in reporting [19,43]. Considering the heterogeneity between studies with pathway, dose and duration, further research is needed to further elucidate this relationship.

A recent review by Kumar et al. [51] highlighted a need for further research involving n-3 PUFA in specific asthma sub-populations, as there is a gap in knowledge for the use of n-3 PUFA in obese subjects with asthma. Considering our findings, which demonstrate that a higher O3I in obese asthmatics is associated with lower CRP and maintenance ICS dose, n-3 PUFA supplementation may provide a unique nonpharmacological approach to treating asthma in this population and demands further research.

As expected, subjects with asthma had poorer lung function and increased airway inflammation compared to those without asthma. Airway inflammation is a key feature of asthma and is characterised by increased levels of eosinophils, neutrophils, or both, in the airways [25,52]. While increased systemic inflammation has been reported in some studies of asthma, this was only significant in CRP between our two groups. This may be related to the higher erythrocyte percentage of n-3 fatty acids that we observed in asthma compared to controls. We are unsure why this occurred in our cohort, but we suspect this must be due to higher dietary intake or supplement use by the asthma group.

Strengths of our study include the use of O3I. O3I is an objective and validated measure of n-3 PUFA intake, as erythrocyte fatty acids represent habitual intake and individual bioavailability as opposed to plasma fatty acids, which reflect shorter term intake [12,22,23]. Erythrocyte fatty acids are more accurate than subjective measures such as FFQs, which rely on accurate recounting of dietary intake, as well as interpretation.

There were some limitations to the study. This was an older cohort, limiting our study’s ability to be generalised to younger populations. In addition, subjects with and without asthma were not matched for sex or BMI. However, where differences existed, analyses have been adjusted for sex, age and BMI. Another limitation was that fish oil supplementation and dietary intake of fish were not recorded across the studies. It would be important to account for this in future research to determine which pathway more effectively changes n-3 PUFA status in this population. Dietary intake data would also be able to address possible confounding by other anti- or proinflammatory foods or nutrients, which we were not able to account for in this study. Lastly, socioeconomic status data for subjects were not available. Considering the established relationship between education and financial status with n-3 PUFA status [53,54] it would have been beneficial to investigate this relationship in a population with asthma, and would be important to assess in future studies. Furthermore, this cohort may be more representative of subjects with high socioeconomic advantage. The higher median O3I in our asthmatic and nonasthmatic subjects was higher than expected for average Australians. Previous research suggests that, on average, Australians consume 395 mg of n-3 PUFA per day [55], equivalent to an O3I of approximately 4–5%.

The cross-sectional study design cannot determine causality; as such, further intervention studies are needed. Nevertheless, our study adds important insight into the relationship between n-3 PUFA and asthma outcomes.

## 5. Conclusions

In conclusion, we have shown that a higher O3I is associated with better asthma control, lower inhaled corticosteroid medication dose and lower systemic inflammatory markers, suggesting that n-3 PUFA may have a role in asthma management. In particular, n-3 PUFA may be clinically relevant for an obese asthma population as our findings show lower ICS dose and CRP in this population with a higher O3I. Our findings suggest that achieving an O3I ≥ 8% may be an appropriate target for therapeutic benefit in both an asthma and an older population. However intervention studies are needed to confirm this hypothesis, particularly in specific subpopulations such as obese people with asthma who may benefit most from this type of dietary intervention.

## Figures and Tables

**Figure 1 nutrients-12-00074-f001:**
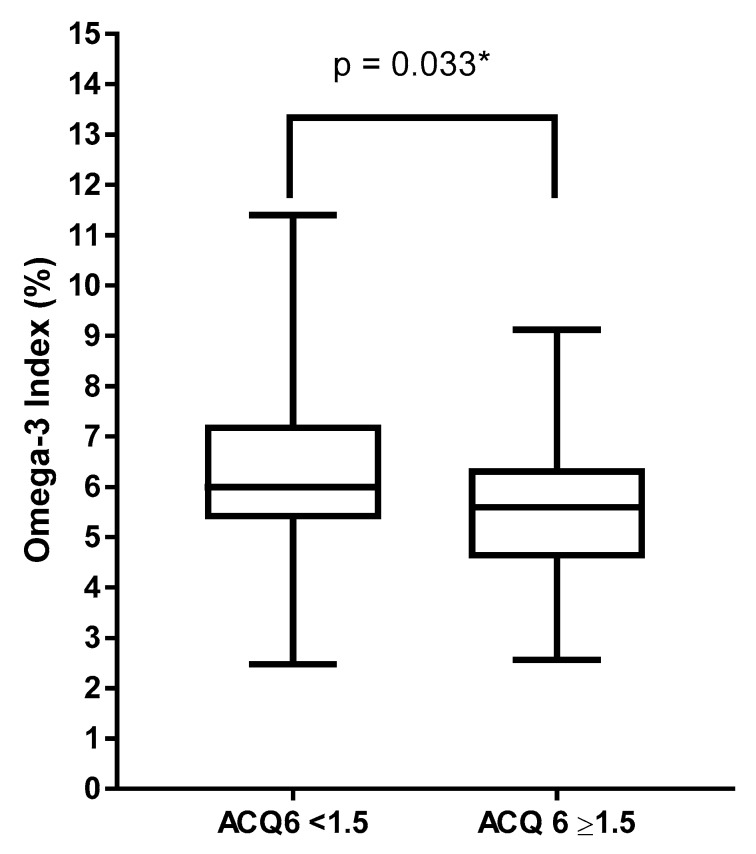
Omega-3 index classified by asthma control, ACQ < 1.5 *n* = 150, ACQ ≥ 1.5 *n* = 44. * Logistic regression adjusting for age, BMI and gender.

**Figure 2 nutrients-12-00074-f002:**
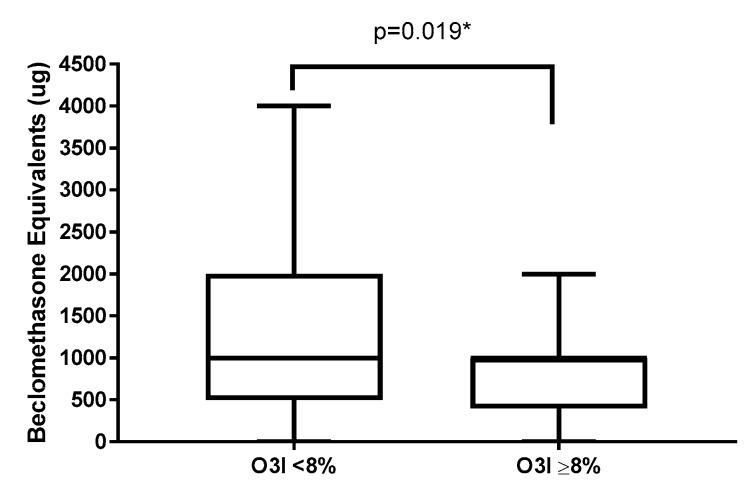
Inhaled corticosteroid dose classified by omega-3 index status. * Logistic regression adjusting for age, BMI and gender.

**Table 1 nutrients-12-00074-t001:** Subject characteristics.

	No Asthma	Asthma	*p*-Value *	Odds Ratio (95% CI)
Subjects	137	255		
Age	53.5 (45.2–64.35)	57.1 (40.9–66.0)	0.8473 ^	
Gender (% female)	39.4 (*n* = 54)	50.6 (*n* = 129)	**0.0435** #	
BMI (kg/m^2^)	33.5 (28.9–41.45)	31.0 (26.9–36.2)	**0.0011** ^	
Smokers (% Ex)	46.72 (*n* = 64)	43.14 (*n* = 110)	0.854	0.98 (0.79–1.22)
Smoking history (pack years)	4.0 (0.0–11.0)	5.5 (0.0–20.0)	**0.011**	1.03 (1.00–1.06)
ACQ6		0.7 (0.2–1.3) (*n* = 206)		
GINA Classification (%1/2/3/4)		25/20/39/16 (*n* = 243)		
FEV_1_ (% predicted)	98.54 ± 13.04 (*n* = 71)	79.42 ± 18.77 (*n* = 235)	**<0.001**	0.93 (0.90–0.95)
FVC (% predicted)	102.8 ± 14.07 (*n* = 71)	91.46 ± 16.11 (*n* = 235)	**<0.001**	0.95 (0.93–0.97)
FEV_1_/FVC (%)	77.0 (74.0–81.0) (*n* = 71)	70.0 (63.1–77.0) (*n* = 235)	**<0.001**	0.88 (0.84–0.92)
Airway markers				
Neutrophils (%)	30.5 (12.0–47.25) (*n* = 51)	34.75 (10.25–54.75) (*n* = 206)	0.782	1.00 (0.99–1.02)
Eosinophils (%)	0.75 (0.25–1.25) (*n* = 51)	2.5 (0.72–14.4) (*n* = 206)	**0.003**	1.25 (1.08–1.44)
Macrophages (%)	61.5 (46.0–74.0) (*n* = 51)	38.31 (18.9–61.31) (*n* = 206)	**<0.001**	0.97 (0.96–0.98)
Lymphocytes (%)	1.75 (0.75–3.0) (*n* = 51)	1.0 (0.25–3.25) (*n* = 206)	**0.032**	1.06 (1.01–1.11)
Systemic markers				
CRP (mg/L)	3.0 (1.5–5.5) (*n* = 135)	3.41 (1.2–8.18) (*n* = 196)	0.018	1.06 (1.01–1.11)
IL-6 (pg/mL)	1.45 (1.05–1.94) (*n* = 34)	1.51 (0.89–2.58) (*n* = 171)	0.289	1.19 (0.87–1.63)
TNF-a (pg/mL)	1.05 (0.88–1.34) (*n* = 44)	1.18 (0.64–1.77) (*n* = 143)	0.168	1.43 (0.86–2.38)
Erythrocyte fatty acids				
SFA (%)	43.29 (42.54–44.26) (*n* = 127)	41.92 (40.87–43.11) (*n* = 242)	**0.001**	0.89 (0.83–0.96)
MUFA (%)	18.27 (17.29–19.28) (*n* = 127)	19.06 (17.67–20.13) (*n* = 242)	0.526	1.02 (0.96–1.09)
PUFA (%)	29.13 (27.27–30.44) (*n* = 127)	28.77 (27.29–30.39) (*n* = 242)	0.995	1.00 (0.91–1.10)
n-3 PUFA (%)	8.91 (7.85–10.25) (*n* = 127)	10.03 (8.74–12.54) (*n* = 242)	**<0.001**	1.24 (1.12–1.37)
O3I (%)	6.1 (4.9–7.4) (*n* = 127)	6.2 (5.4–7.9) (*n* = 242)	0.089	1.10 (0.99–1.23)
n-6:n-3	3.31 (2.72–3.74) (*n* = 127)	2.82 (2.30–3.40) (*n* = 242)	**<0.001**	0.58 (0.44–0.78)

Data are presented as median (interquartile range) or mean ± standard deviation. Significant effects are highlighted in bold. BMI: Body mass index; FEV_1_: Forced expiratory volume in 1 s; FVC: Forced vital capacity; ACQ: Asthma Control Questionnaire; GINA: Global Initiative for Asthma; CRP: C-reactive protein; IL-6: Interleukin-6; TNF-α: Tumour necrosis factor α; SFA: Saturated fatty acids; MUFA: Monounsaturated fatty acids; n-6 PUFA: Omega-6 polyunsaturated fatty acids; n-3 PUFA: Omega-3 polyunsaturated fatty acids; O3I: Omega-3 index. GINA classification: 1 = intermittent, 2 = mild persistent, 3 = moderate persistent, 4 = severe persistent. *: Logistic regression analysis performed, adjusting for age, gender and BMI unless otherwise stated. Reference population is nonasthmatic. ^: Mann–Whitney test, unadjusted. #: Chi-squared test, unadjusted.

**Table 2 nutrients-12-00074-t002:** Clinical asthma markers of subjects with asthma by O3I status.

	Asthma, Low O3I (<8%)	Asthma, High O3I (≥8%)	*p*-Value *	Odds Ratio (95% CI)
*n*	185	57		
Age	57.9 (42.8–66.4)	54.6 (36.5–65.6)	0.1938 ^	
BMI (kg/m^2^)	30.6 (26.9–37.7)	31.7 (28.1–35.5)	0.9591 ^	
ICS (ug beclomethasone eq/d)	1000 (500–2000) (*n* = 120)	1000 (400–1000) (*n* = 51)	**0.019**	0.999 (0.9989–0.9999)
ACQ6	0.7 (0.2–1.3) (*n* = 170)	0.42 (0.2–1.11) (*n* = 24)	0.311	0.71 (0.37–1.37)
GINA Classification: 1/2/3/4 (%)	24/22/38/16 (*n* = 176)	19/25/41/15 (*n* = 54)	0.7638 #	
FEV_1_ (% predicted)	79.11 ± 19.4 (*n* = 169)	78.93 ± 16.78 (*n* = 54)	0.610	1.00 (0.98–1.01)
FVC (% predicted)	91.01 ± 16.82 (*n* = 169)	91.71 ± 13.43 (*n* = 54)	0.760	1.00 (0.98–1.02)
FEV_1_/FVC (%)	70 (63–77) (*n* = 169)	69.55 (59.78–75.33) (*n* = 54)	0.249	0.98 (0.95–1.01)
CRP (mg/mL)	3.8 (1.3–8.7) (*n* = 140)	2.9 (1.1–7.8) (*n* = 44)	0.124	0.94 (0.87–1.02)
IL-6 (pg/mL)	1.8 (1.3–8.7) (*n* = 41)	0.9 (0.1–1.7) (*n* = 41)	0.212	0.85 (0.65–1.10)
TNF (pg/mL)	1.3 (0.9–1.8) (*n* = 99)	0.4 (0.2–1.2) (*n* = 37)	0.944	1.01 (0.85–1.19)

Data are presented as median (interquartile range) or mean ± standard deviation unless stated. Significant effects are highlighted in bold. ICS: inhaled corticosteroid; ACQ: Asthma Control Questionnaire; GINA: Global Initiative for Asthma; FEV_1_: Forced expiratory volume in 1 s; %; FVC: Forced vital capacity; CRP: C-reactive protein; IL-6: Interleukin-6; TNF-α: Tumour necrosis factor α. *: Logistic regression analysis performed, adjusting for age, gender and BMI unless otherwise stated. Reference population is low O3I. ^: Mann–Whitney test, unadjusted. #: Chi-squared test, unadjusted.

**Table 3 nutrients-12-00074-t003:** Clinical asthma markers and inflammatory markers of subjects with asthma, classified by weight and O3I.

	Obese Asthma		Nonobese Asthma					
Obese Asthma	>8% O3I	<8% O3I	>8% O3I	<8% O3I	*p*-Value *	Obesity × O3I Interaction	High vs. Low O3I	Obese vs. Nonobese
*n*	37	98	20	87				
ICS (μg beclomethasone eq/d)	1000 (212.5–1000) (*n* = 36)	1000 (1000–2000) (*n* = 61)	1000 (500–1000) (*n* = 15)	500 (500–1000) (*n* = 59)	**0.0002**	0.1556	0.0104	0.0077
ACQ6	0.33 (0–1.8) (*n* = 7)	0.75 (0.3–1.5) (*n* = 86)	0.5 (0.3–1.0) (*n* = 17)	0.7 (0.2–1.3) (*n* = 84)	0.6981	0.8864	0.4235	0.4948
FEV_1_ (% predicted)	82.49 ± 15.08 (*n* = 35)	80.18 ± 19.37 (*n* = 84)	72.37 ± 18.15 (*n* = 19)	78.05 ± 19.48 (*n* = 85)	**0.0001**	0.6616	0.4851	0.3445
FVC (% predicted)	94.04 ± 13.87 (*n* = 35)	88.61 ± 15.84 (*n* = 84)	87.42 ± 11.74 (*n* = 19)	93.38 ± 17 (*n* = 85)	**<0.0001**	0.2197	0.8011	0.3537
FEV_1_/FVC (5)	71.5 (63.9–77.7) (*n* = 35)	74 (66.25–80) (*n* = 84)	66 (55–71.6) (*n* = 19)	68 (60–74) (*n* = 85)	**<0.0001**	0.6012	0.1146	0.0035
CRP (mg/mL)	3.1 (1.25–8) (*n* = 33)	5.59 (2.67–12.42) (*n* = 74)	1.57 (0.9–7.28) (*n* = 11)	1.77 (0.97–5) (*n* = 67)	**0.0003**	0.0475		
IL-6 (pg/mL)	0.30 (0.04–1.13) (*n* = 30)	2.35 (1.61–3.37) (*n* = 53)	1.77 (1.16–2.42) (*n* = 11)	1.38 (0.9–2.11) (*n* = 65)	**0.0326**	0.6909	0.1857	0.3207
TNF (pg/mL)	0.35 (0.18–0.9) (*n* = 27)	1.41 (0.96–1.92) (*n* = 50)	1 (0.46–1.54) (*n* = 10)	1.23 (0.82–1.75) (*n* = 49)	0.4769	0.3315	0.8142	0.2666

Data are presented as median (interquartile range) or mean ± standard deviation. Significant effects are highlighted in bold. ICS: inhaled corticosteroid; ACQ: Asthma Control Questionnaire; GINA: Global Initiative for Asthma; FEV1: Forced expiratory volume in 1 s; FVC: Forced vital capacity; CRP: C-reactive protein; IL-6: Interleukin-6; TNF-α: Tumour necrosis factor α. *: Two-factor ANOVA analysis performed, adjusted for age and gender, unless otherwise stated.

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
