# Peer review of "Higher Omega-3 Index Is Associated with Better Asthma Control and Lower Medication Dose: A Cross-Sectional Study"

_nutrients, 2019, doi:10.3390/nu12010074_

Round 1

Reviewer 1 Report

General comments

The role of diet in asthma in an important topic for research. The study measured erythrocyte omega-3 polyunsaturated fatty acids (n-3 PUFA) in adults with asthma (n=255) compared with a non-asthma control group (n=137) and the association of n-3 PUFA levels with clinical asthma outcomes. The main findings were that a low omega-3 index (O3I) was associated with worse asthma control and higher ICS dose. The authors speculate that higher erythrocyte n-3 PUFA level may have a role in asthma management. The results are new and of interest. The study has been well performed and the methods seem to be appropriate. Limitations of the study are acknowledged including older age of the participants, some mismatch in demographics between groups (gender, BMI) and lack of information of dietary intake of groups for fish and fish oil supplementation. I’ve a few specific comments that are listed below

Major comments

Social deprivation

If available, data on social deprivation of the participants should be included in the paper. Differences between participants in social deprivation index could influence omega-3 index (O3I) levels

Pack year history

Table 1: To provide an index on the intensity of exposure to tobacco smoke include pack year history

Author Response

Title: Higher omega-3 index is associated with better asthma control and lower medication dose: a cross sectional study

Manuscript ID: nutrients-643430

The role of diet in asthma in an important topic for research. The study measured erythrocyte omega-3 polyunsaturated fatty acids (n-3 PUFA) in adults with asthma (n=255) compared with a non-asthma control group (n=137) and the association of n-3 PUFA levels with clinical asthma outcomes. The main findings were that a low omega-3 index (O3I) was associated with worse asthma control and higher ICS dose. The authors speculate that higher erythrocyte n-3 PUFA level may have a role in asthma management. The results are new and of interest. The study has been well performed and the methods seem to be appropriate. Limitations of the study are acknowledged including older age of the participants, some mismatch in demographics between groups (gender, BMI) and lack of information of dietary intake of groups for fish and fish oil supplementation. I’ve a few specific comments that are listed below.

Response: Thank you for your time reviewing the paper, and for your feedback.

Major comments

Comment 1: Social deprivation

If available, data on social deprivation of the participants should be included in the paper. Differences between participants in social deprivation index could influence omega-3 index (O3I) levels

Response: Unfortunately we do not have any socioeconomic data on our subjects. The authors agree this is an important point, and a limitation of the study. This may account for the relatively high O3I in both the asthma and non-asthma group as previous literature suggests the average Australian population has an index of about 4%. We have included this in our limitations section of the paper.

Discussion (line 353-360): “Lastly, socioeconomic status data for subjects was not available. Considering the established relationship between education and financial status with n-3 PUFA status [53,54] it would have been beneficial to investigate this relationship in a population with asthma, and would be important to assess in future studies. Furthermore, this cohort may be more representative of subjects with high socioeconomic advantage. The higher median O3I in our asthma and non-asthma subjects was higher than expected for average Australians. Previous research suggests that on average Australians consume 395mg of n-3 PUFA per day [55], equivalent to an O3I of approximately ~4-5%.”

Comment 2: Pack year history

Table 1: To provide an index on the intensity of exposure to tobacco smoke include pack year history

Response: We have included a smoking history variable in pack years in Table 1.  (line 197)

Reviewer 2 Report

The paper of Stoodley et al aimed at investigating if the omega-3 index is associated with better asthma control. The study has some main issues:

1) Despite the Introduction is well written, it is not clear why the authors wanted to investigate the effects of obesity on O3I in their population: lack a rationale and a hypothesis.

2) If the aim of the study is the one above enunciated, it is not clear why they included in the study patients without asthma. Similarly, the table 1 in not informative: what the reader would like to see is the difference between low and high O3I.

3) The inclusion and exclusion criteria and more details about the study design should be reported. Were the tests all performed at the same day visit? Were they available for all patients?

4) Pearson’s test doesn’t study the association between two variables; furthermore, it is not clear what this statistic is used for.

5) The authors stated that there is a “statistically significant difference”/”significant difference” in ICS doses between patients with lower and higher O3I, but reported the same median value: 1000 and 1000. Despite the statistically significant (?) difference, there is no difference between the values reported as median.

Author Response

Title: Higher omega-3 index is associated with better asthma control and lower medication dose: a cross sectional study

Manuscript ID: nutrients-643430

The paper of Stoodley et al aimed at investigating if the omega-3 index is associated with better asthma control. The study has some main issues:

Comment 1:

Despite the Introduction is well written, it is not clear why the authors wanted to investigate the effects of obesity on O3I in their population: lack a rationale and a hypothesis.

Response: Thank you for your feedback and taking the time to review our paper. We have reworded the introduction to include a rational and hypothesis for investigating the effects of O3I and obesity on asthma outcomes.

Introduction (line 79-97): “The conflicting evidence highlights a need for more research in this area. Hence, this study aimed to examine the relationship between n-3 PUFA status and clinical outcomes in Australian adults with asthma. Firstly, it is unclear whether n-3 PUFA status is impaired in Australian subjects with asthma compared to health controls, thus an aim of this project was to investigate and describe the differences between these two groups. We hypothesised that individuals with asthma would have poorer n-3 PUFA status compared to those without asthma. Furthermore we hypothesized that subjects with asthma and a high n-3 PUFA status would have better clinical outcomes than those with low n-3 PUFA status. These aims were examined using the omega-3 index (O3I), which has been validated as a reliable measure of dietary n-3 PUFA intake and reflects long-term n-3 PUFA status [20,21]. O3I is the sum of erythrocyte EPA and DHA, expressed as a percentage of total erythrocyte membrane fatty acids [22]. A secondary aim of this project was to examine the effects of obesity on O3I in adults with asthma. Obesity in asthma is associated with poorer asthma control, greater severity, higher medication doses and more frequent exacerbations than healthy weight individuals [Stoodley]. One mechanism suggested to underpin this relationship is the chronic low grade inflammation associated with obesity [Wood, Scott].  Considering the anti-inflammatory properties attributed to n-3 PUFA, it is possible that n-3 PUFA may attenuate this inflammation.  Whether these interactions exist in obese asthmatic subjects is unknown. Therefore we hypothesized that in an obese asthmatic population, those with a lower O3I would have poorer clinical and biochemical outcomes compared to those with a higher O3I.”

Comment 2

If the aim of the study is the one above enunciated, it is not clear why they included in the study patients without asthma. Similarly, the table 1 in not informative: what the reader would like to see is the difference between low and high O3I.

Response: Please see the above changes in comment 1 to the rationale and hypothesis. The authors believe it is important to investigate O3I and clinical outcomes in both populations to give context to the results in the asthma only population. In addition, this is the first Australian study to investigate erthyrocyte fatty acid status in subjects with and without asthma. We have therefore kept Table 1 and included a rationale and hypothesis. (see above changes to Introduction).

Comment 3

The inclusion and exclusion criteria and more details about the study design should be reported. Were the tests all performed at the same day visit? Were they available for all patients?

Response: Clinical assessment and blood collection were performed during a single research visit for all participants in the study. We have specified this in the methods section.

Methods (line 114):“Clinical assessment and blood collection were performed during a single clinic visit.”

Response: Not all tests were performed in all participants, however the number of subjects has been reported in each results table for variables where data was not available for the whole group. Inclusion and exclusion criteria is described in paragraph 2.1 Subjects. We have added further information about exclusion criteria.

Methods (line 107-108): “Exclusions included current smokers, use of systemic anti-inflammatory or immunosuppressant medications or current cancer diagnosis.”

Comment 4

Pearson’s test doesn’t study the association between two variables; furthermore, it is not clear what this statistic is used for.

Response: Thank you for picking this up! We do not present associations in this paper, this has been removed.

Comment 5

The authors stated that there is a “statistically significant difference”/”significant difference” in ICS doses between patients with lower and higher O3I, but reported the same median value: 1000 and 1000. Despite the statistically significant (?) difference, there is no difference between the values reported as median.

Response: As the ICS dose variable  was non-parametric, the authors presented the data as median with interquartile range. We used the Mann Whitney test to test for differences between the two groups for the same reason. The Mann Whitney test ranks all values from low to high, and compares the means of these ranks, rather than directly testing the medians. Therefore, this test can detect differences in shape and spread between the two groups.  It is consequently possible to have a statistically significant difference between the two groups while displaying an identical median. This spread difference is the clinically significant result we detected between our groups. We found that individuals with a higher O3I had a smaller spread of ICS maintenance dose, and those with a lower O3I had a greater range of ICS maintenance dose. Given this confusion, we have clarified this difference in our paper. We have included a reference for this rationale below. 

http://www.bmj.com/cgi/content/full/323/7309/391

Results (line 207-211): “When analysing clinical asthma measures, there was no significant difference in lung function in subjects with a higher (≥8%) versus lower (<8%) O3I (Table 2). However subjects with higher O3I had a significantly lower range of maintenance ICS dose (beclomethasone equivalents) (Table 2, Figure 2).”

Results (line 223-225): “We found that subjects who were obese with a lower O3I had a significantly higher range of maintenance ICS medication dose compared with obese subjects with a higher O3I (p=0.0002) (Table 3).”

Discussion (lines 266-267): “Another significant finding was the relationship between O3I and maintenance ICS dose. Subjects with a higher O3I had a significantly lower range of maintenance ICS dose.”

Discussion (lines 288-291): “When we examined obese and non-obese asthmatics according to O3I, obese asthmatics with a higher O3I had a lower range of maintenance ICS doses compared with obese asthmatics with a lower O3I. This is particularly important, as obesity is associated with a reduced response to ICS medication, requiring higher doses to achieve protective effects [43].”

Reviewer 3 Report

This study investigates the association between the omega-3 index and asthma outcomes in Australian adults. Although some interesting and consistent results were found, there are in my opinion some serious limitations (some of which have not been acknowledged in the Discussion) that, in my opinion, make it very difficult to rule out confounding and bias.

Specific suggestions and comments are listed below.

Apart from the cross-sectional nature of the study, the major weakness of this study is, in my opinion, the lack of adjustment for many major potential confounders. Why were the analyses not multivariate? Results should at least be adjusted for age, BMI and all available potential confounders. Although this was not discussed in the limitations, I assume there was no available information on e.g. SES factors? Given how both dietary habits and asthma outcomes (severity, control, medication use) are strongly correlated with social, environmental and lifestyle factors, it is in my opinion impossible to rule out confounding as an explanation for the associations found between omega-3 index and asthma outcomes in this observational study.

Following-up on my previous comment, although the use of O3I as an objective and validated measure of n-3 PUFA intake is indeed a strength of the study, given the lack of information on other (objective or estimated) dietary exposure, it is again difficult to rule out potential confounding by other (anti-inflammatory or more generally healthy) foods or nutrients.

The strengths and limitations section lacks discussion in my opinion. Moreover, there are some major limitations that in my opinion have not been acknowledged (as mentioned above) and some strengths that I would not have cited as such (e.g. the sample size).

The data available on ICS use has not been described in the Methods. Moreover, I am not sure I understand why the authors considered ICS as a continuous variable: were all asthmatic subjects using ICS? If not, I suggest considering ICS use as a binary variable (yes/no) and compare the proportion of ICS use among subjects with a high versus low O3I.

I think that the aim of the study (i.e. to examine the relationship between n-3 PUFA status and clinical outcomes in Australian adults with asthma) does not match the rest of the paper as some analyses were made to compare adults with asthma to adults without asthma. Please clarify.

Author Response

Title: Higher omega-3 index is associated with better asthma control and lower medication dose: a cross sectional study

Manuscript ID: nutrients-643430

This study investigates the association between the omega-3 index and asthma outcomes in Australian adults. Although some interesting and consistent results were found, there are in my opinion some serious limitations (some of which have not been acknowledged in the Discussion) that, in my opinion, make it very difficult to rule out confounding and bias.

Specific suggestions and comments are listed below.

Comment 1:

Apart from the cross-sectional nature of the study, the major weakness of this study is, in my opinion, the lack of adjustment for many major potential confounders. Why were the analyses not multivariate? Results should at least be adjusted for age, BMI and all available potential confounders. Although this was not discussed in the limitations, I assume there was no available information on e.g. SES factors? Given how both dietary habits and asthma outcomes (severity, control, medication use) are strongly correlated with social, environmental and lifestyle factors, it is in my opinion impossible to rule out confounding as an explanation for the associations found between omega-3 index and asthma outcomes in this observational study.

 Response: Thank you for taking the time to review our paper. The authors agree there are many confounders associated with O3I, and unfortunately data for some potential confounding factors was not available. In particular, we did not have data for socioeconomic status which would have been interesting to investigate, particularly in an asthma population. We have added this limitation into our discussion.

Discussion (lines 346-360): “There were some limitations to the study. This was an older cohort, limiting our study’s ability to be generalised to younger populations. In addition, subjects with and without asthma were not matched for sex or BMI. However, where differences existed, analyses have been adjusted for sex, age and BMI. Another limitation was that fish oil supplementation and dietary intake of fish were not recorded across the studies. It would be important to account for this in future research to determine which pathway more effectively changes n-3 PUFA status in this population. Dietary intake data would also be able to address possible confounding by other anti- or pro-inflammatory foods or nutrients, which we were not able to account for in this study. Lastly, socioeconomic status data for subjects was not available. Considering the established relationship between education and financial status with n-3 PUFA status [53,54] it would have been beneficial to investigate this relationship in a population with asthma, and would be important to assess in future studies. Furthermore, this cohort may be more representative of subjects with high socioeconomic advantage. The higher median O3I in our asthma and non-asthma subjects was higher than expected for average Australians. Previous research suggests that on average Australians consume 395mg of n-3 PUFA per day [55], equivalent to an O3I of approximately ~4-5%.”

Response: For the comparison between low and high O3I in our asthmatic population, we have included both age and BMI within the table. We have also adjusted Table 1 and 2 results based on age, gender and BMI. This has changed some of our results, in particular the differences in systemic inflammation are no longer significant (we have hence removed Figure 3). Please see revised Table 1 and 2.

Results (line 207-211): “When analysing clinical asthma measures, there was no significant difference in lung function or asthma severity in subjects with a higher (≥8%) versus lower (<8%) O3I (Table 2). However subjects with higher O3I had a significantly lower range of maintenance ICS dose (beclomethasone equivalents) (Table 2, Figure 2). There were no significant differences in systemic inflammatory markers between higher and lower O3I after adjusting for age, gender and BMI (Table 2).”

Response: In our last table, while we accounted for BMI by comparing obese and non-obese we found there was a significant difference between the groups for age (we found that the obese participants with a high O3I were slightly younger than the participants in both non-obese groups). As such, we have performed the analysis adjusting for age and gender, and updated the table. Please see the revised Table 3, with nil significant changes.

Results (lines 219-227): “Due to the interesting nexus of asthma and obesity, we compared clinical asthma markers and systemic inflammation in subjects with asthma divided into obese (BMI ≥30kg/m2) and non-obese (BMI <30kg/m2) groups with high (≥8%) and low (<8%) O3I (Table 3). The analysis was adjusted for gender and age. We found that subjects who were obese with a lower O3I had a significantly higher range of maintenance ICS medication dose compared with obese subjects with a higher O3I (p=0.0002) (Table 3). While there was significance difference detected in all lung function measures, only FEV1/FVC ratio revealed an obesity interaction. Both obesity and O3I were significant predictors of CRP, but not for TNF-α or Il-6. Obesity and O3I were not significant predictors for ACQ.”

Comment 2

Following-up on my previous comment, although the use of O3I as an objective and validated measure of n-3 PUFA intake is indeed a strength of the study, given the lack of information on other (objective or estimated) dietary exposure, it is again difficult to rule out potential confounding by other (anti-inflammatory or more generally healthy) foods or nutrients.

 Response: The authors agree that it is a limitation of our work that we did not collect dietary or supplement data on our participants, and highlight this should be examined in future work. However, we believe the strength of O3I is in its reflection of bioavailable omega-3 fatty acids. Being able to provide a recommendation on the bioavailable levels needed to address inflammation in asthma allows future studies to investigate both supplement and dietary intakes of fatty acids, or potentially a combined effect. While it would be interesting to know whether supplementation or food sources altered levels, we feel we adequately answer our question, which is on status, rather than doses or pathways required.

Comment 3

The strengths and limitations section lacks discussion in my opinion. Moreover, there are some major limitations that in my opinion have not been acknowledged (as mentioned above) and some strengths that I would not have cited as such (e.g. the sample size).

 Response: We have made significant changes to this section of the paper. Please see the response to comment 1 which details the changes to the strengths and limitations section. We have also removed sample size as a strength.

Comment 4

The data available on ICS use has not been described in the Methods. Moreover, I am not sure I understand why the authors considered ICS as a continuous variable: were all asthmatic subjects using ICS? If not, I suggest considering ICS use as a binary variable (yes/no) and compare the proportion of ICS use among subjects with a high versus low O3I.

 Response: We believe there is some confusion over this outcome of our paper. We are investigating maintenance ICS dose rather than use. While we agree use would be a categorical variable the dose of ICS used by participants is continuous. We have added in the methods how ICS dose was recorded and clarified throughout the paper that dose was being assessed rather than use.

Methods (lines 121-122): “Maintenance ICS doses were recorded and converted to beclomethasone equivalents.”

Comment 5

I think that the aim of the study (i.e. to examine the relationship between n-3 PUFA status and clinical outcomes in Australian adults with asthma) does not match the rest of the paper as some analyses were made to compare adults with asthma to adults without asthma. Please clarify.

 Response: The authors agree the aims and hypotheses do not match up with the entirety of the paper. Please see below changes to the introduction specifying the various aims.

Introduction (line 79-97): “The conflicting evidence highlights a need for more research in this area. Hence, this study aimed to examine the relationship between n-3 PUFA status and clinical outcomes in Australian adults with asthma. Firstly, it is unclear whether n-3 PUFA status is impaired in Australian subjects with asthma compared to health controls, thus an aim of this project was to investigate and describe the differences between these two groups. We hypothesized that individuals with asthma would have poorer n-3 PUFA status compared to those without asthma. Furthermore we hypothesized that subjects with asthma and a high n-3 PUFA status would have better clinical outcomes than those with low n-3 PUFA status. These aims were examined using the omega-3 index (O3I), which has been validated as a reliable measure of dietary n-3 PUFA intake and reflects long-term n-3 PUFA status [21,22]. O3I is the sum of erythrocyte EPA and DHA, expressed as a percentage of total erythrocyte membrane fatty acids [23]. A secondary aim of this project was to examine the effects of obesity on O3I in adults with asthma. Obesity in asthma is associated with poorer asthma control, greater severity, higher medication doses and more frequent exacerbations than healthy weight individuals [24]. One mechanism suggested to underpin this relationship is the chronic low grade inflammation associated with obesity[25,26].  Considering the anti-inflammatory properties attributed to n-3 PUFA, it is possible that n-3 PUFA may attenuate this inflammation. Whether these interactions exist in obese asthmatic subjects is unknown. Therefore we hypothesized that in an obese asthmatic population, those with a lower O3I would have poorer clinical and biochemical outcomes compared to those with a higher O3I.”

Round 2

Reviewer 1 Report

The Authors have adequately addressed my comments

Reviewer 2 Report

The authors have addressed all my comments.

Reviewer 3 Report

I thank the authors for sending this revised version of their manuscript. I think that my comments have  properly been addressed and that the revised manuscript has significantly been improved.